# Using genetic variation to disentangle the complex relationship between food intake and health outcomes

Nicola Pirastu[1,2]*, Ciara McDonnell[1,3], Eryk J. Grzeszkowiak[1], Ninon Mounier[4,5,6], Fumiaki Imamura[7], Jordi Merino[8,9,10], Felix R. Day[7], Jie Zheng[11], Nele Taba[13,14,15], Maria Pina Concas[14], Linda Repetto[1], Katherine A. Kentistou[1,3], Antonietta Robino[14], Tõnu Esko[15], Peter K. Joshi[1], Krista Fischer[12,15], Ken K. Ong[7], Tom R. Gaunt[11], Zoltán Kutalik[4,5,6], John R. B. Perry[7], James F. Wilson[1,16]

1 Centre for Global Health Research, Usher Institute, University of Edinburgh, Edinburgh, Scotland, United Kingdom, 2 Human Technopole, Milan, Italy, 3 Centre for Cardiovascular Sciences, Queen's Medical Research Institute, University of Edinburgh, Edinburgh, Scotland, United Kingdom, 4 Centre for Primary Care and Public Health, University of Lausanne, Lausanne, Switzerland, 5 Department of Computational Biology, University of Lausanne, Lausanne, Switzerland, 6 Swiss Institute of Bioinformatics, Lausanne, Switzerland, 7 MRC Epidemiology Unit, Institute of Metabolic Science, University of Cambridge School of Clinical Medicine, Cambridge, United Kingdom, 8 Diabetes Unit and Centre for Genomic Medicine, Massachusetts General Hospital, Boston, Massachusetts, United States of America, 9 Program in Medical and Population Genetics, Broad Institute, Cambridge, Massachusetts, United States of America, 10 Department of Medicine, Harvard Medical School, Boston, Massachusetts, United States of America, 11 MRC Integrative Epidemiology Unit, Bristol Medical School, Bristol, United Kingdom, 12 Institute of Mathematics and Statistics, University of Tartu, Tartu, Estonia, 13 Institute of Molecular and Cell Biology, University of Tartu, Tartu, Estonia, 14 Institute for Maternal and Child Health—IRCCS "Burlo Garofolo", Trieste, Italy, 15 Estonian Genome Centre, Institute of Genomics, University of Tartu, Tartu, Estonia, 16 MRC Human Genetics Unit, Institute of Genetics and Cancer, University of Edinburgh, Edinburgh, Scotland, United Kingdom

☯ These authors contributed equally to this work.
* nicola.pirastu@fht.org

**Data Availability Statement:** Full GWAS results are available through GWAS catalog accession ID

## Abstract

Diet is considered as one of the most important modifiable factors influencing human health, but efforts to identify foods or dietary patterns associated with health outcomes often suffer from biases, confounding, and reverse causation. Applying Mendelian randomization in this context may provide evidence to strengthen causality in nutrition research. To this end, we first identified 283 genetic markers associated with dietary intake in 445,779 UK Biobank participants. We then converted these associations into direct genetic effects on food exposures by adjusting them for effects mediated via other traits. The SNPs which did not show evidence of mediation were then used for MR, assessing the association between genetically predicted food choices and other risk factors, health outcomes. We show that using all associated SNPs without omitting those which show evidence of mediation, leads to biases in downstream analyses (genetic correlations, causal inference), similar to those present in observational studies. However, MR analyses using SNPs which have only a direct effect on the exposure on food exposures provided unequivocal evidence of causal associations between specific eating patterns and obesity, blood lipid status, and several other risk factors and health outcomes.

GCP000298 (http://ftp.ebi.ac.uk/pub/databases/gwas/summary_statistics/GCST90096001-GCST90097000/GCST90096892/) All results from the MR analyses have been shared in the additional tables.

**Funding:** J.F.W. acknowledges support from the MRC Human Genetics Unit quinquennial programme grant "QTL in Health and Disease" (MC_UU_00007/10). EGCUT was funded by Estonian Research Council Grant IUT20-60, PRG1291 (T.E.), PUT1665 (K.F.), PRG1197 (K.F.), the European Union through the European Regional Development Fund grant no. 2014-2020.4.01.15-0012 GENTRANSMED (T.E.) and 2014-2020.4.2.2 (N.T.) and 2014-2020.4.2.2(T.E.), and Estonian and European Research Roadmap grant no.2014-2020.4.01.16-0125(T.E.). The EPIC-Norfolk study (DOI 10.22025/2019.10.105.00004) has received funding from the Medical Research Council MR/N003284/1, MCPC_13048, MC-UU_12015/1, and MC_UU_00006/1 (J.P., K.O). The Fenland study (DOI: 10.1186/ISRCTN72077169) was funded by the Medical Research Council and the Wellcome Trust Ref: 074548.( J.P., K.O., F.I. and F.R.D). J.P., K.O., F.I. and F.R.D were funded by the UK Medical Research Council Epidemiology Unit core grant MC_UU_00006/2 and MC_UU_00006/3. T.R.G. receives funding from the UK Medical Research Council (MC_UU_00011/4). Z.K. received funding from the Swiss National Science Foundation (31003A_169929). J.M. was partially supported by American Diabetes Association grant #7-21-JDFM-005 and by the National Institutes of Health grant P30 DK040561. The funders had no role in study design, data collection and analysis, decision to publish, or preparation of the manuscript.

**Competing interests:** I have read the journal's policy and the authors of this manuscript have the following competing interests: Dr Joshi is a paid consultant to Global Gene Corp and Humanity Inc.

## Author summary

Food and drink consumption is one of the most important factors influencing human health and wellbeing. The role of diet in human physiology and disease has been widely studied, but challenges in accurately assessing long term diet result in contradicting findings. Mendelian randomization is a statistical technique that uses genetic variants associated with modifiable exposures to estimate the causal effect of an exposure to a health outcome and could be extremely useful in the context of diet-health relationships. In our study, we initially identified genetic variants associated to 29 measures of food and drink consumption. We then show that genetic variants associated with food and drink consumption are subject to reverse causation and confounding. We have thus developed a statistical genetics method to identify genetic variants directly associated with food and drink consumption. By using these genetic variants (and their corresponding direct effects) in Mendelian randomization analyses we provided consistent evidence of causal associations of food and drink consumption with obesity, blood lipid status, and several other risk factors and health outcomes.

## Introduction

Given its impact on human well-being, diet is one of the most studied human behaviours. Quality, quantity, and patterns of consumed foods are associated with a wide range of medical conditions such as metabolic, inflammatory, or mental health diseases.[1] However, despite the growing number of studies reporting associations between diet and health outcomes, it has been challenging to establish causal relationships due methodological limitations such as measurement error, confounding, and reverse causation [2,3]. To date, several approaches have been devised to try to account for intrinsic limitations in nutritional studies such as the use of methods to calibrate food records [4] through the use of 24h recalls [5], biomarkers [6] and doubly labelled water, [7] or the implementation of domiciled feeding studies.[8] Although the implementation of these methods or study designs have helped in addressing some of the limitations of nutrition research, difficulties remain especially when it comes to estimate the causal effect of diet on health outcomes.

In this context genetics may represent an alternative approach by the use of Mendelian Randomization (MR). MR is a methodological approach in which genetic variants associated with an exposure of interest are used as instrumental variables to investigate the causal association between this exposure and an outcome.[9] To date, several MR studies have been designed to investigate the associations between the consumption of single food groups, such as alcoholic beverages [10], coffee [11], milk [12–14] and specific health outcomes, but a systematic study investigating the overall role of diet on multiple health outcomes is missing. Previous MR studies have not accounted for the fact that genetic variants associated with reported dietary intake may be primarily associated with other risk factors, reporting characteristics or social determinants of health which may confound the causal estimates.

The present study was designed to initially identify the genetic variants associated with reported food consumption, and then to leverage a causal inference statistical framework to systematically investigate the causal effects of dietary factors on health outcomes while accounting for the reverse causal effects that health determinants have on habitual dietary intake reporting.

## Methods

Given large number of analyses conducted for this study and their complexity Fig 1 summarises the main analyses.

### Study population and genome-wide association for dietary intake

The UK Biobank [15] is a large population-based cohort including 500 000 adults aged between 40 and 69 years at baseline across 22 assessments centres in the United Kingdom. Data were collected based on clinical examinations, assays of biological samples, detailed information on self-reported health characteristics, and genome-wide genotyping. Dietary intake in UK Biobank was assessed using a touchscreen dietary frequency questionnaire which included questions about the frequency of consumption specific foods and beverages over the past year. The number of samples used for each trait can be found in Table A in S1 Table while a detailed description of the phenotypes, can be found in the in the S1 Note 1.2 and Table B in S1 Table. For alcohol consumption traits analyses were limited to people drinking at least one glass of the alcoholic beverage a week. This choice was due to very high number of people who reported to drink 0 glasses of the specific alcoholic beverage per week (between 58% for red wine to 94% for fortified wine) which would have biases the analyses, due to reverse causation [16]. A similar issue applies to coffee consumption traits where stratifying for coffee type required to restrict the analysis to coffee drinkers. Finally, we have excluded people who reported eating certain foods (e.g. beef) less than once a week due to the very large range of different consumptions which this response corresponds to. The proportion of people used for the analysis compared to the overall UK biobank participants can be found in Table A in S1 Table. Validity of the food consumption measures has been evaluated by Bradbury and others

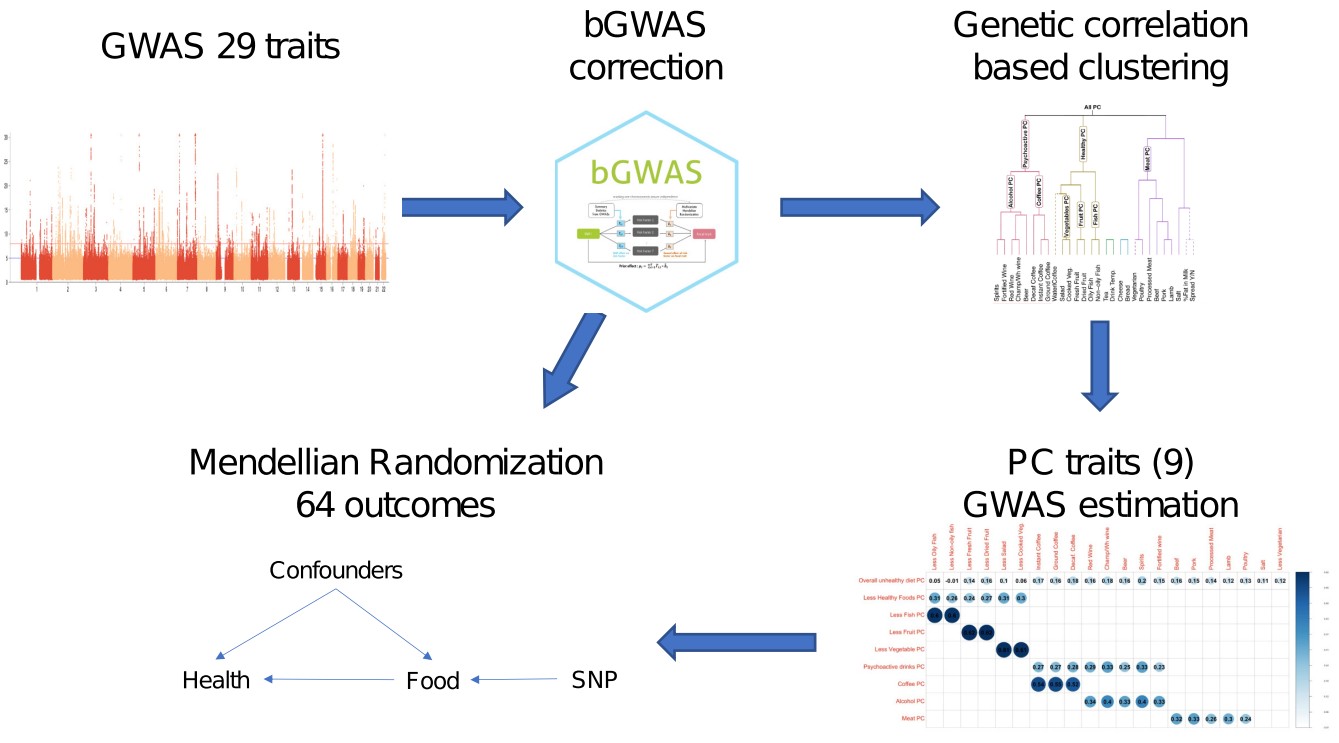

**Fig 1. Overall study design.**

[17] which concluded that "the dietary touchscreen variables, available on the full cohort, reliably rank participants according to intakes of the main food groups".

We used the BOLT-LMM software [18] to assess the association between the genetic variants across the human genome and 29 food phenotypes. Analyses were conducted on genetic data release version 3 imputed to the HRC panel [19], as provided by the UK Biobank (http://www.ukbiobank.ac.uk/wp-content/uploads/2014/04/UKBiobank_genotyping_QC_documentation-web.pdf). Population stratification was assessed using LD-score regression as implemented in ldsc [20,21] using the LD scores provided with the software which refer to the HapMap [22] v3 SNPs. Table Q in S1 Table reports for each food trait the LD regression intercept and heritability estimation using the ldsc software [20]. Cluster analysis conducted on the foods identified five main independent groups of traits (see additional online methods paragraph 1.8 and 2.2 for details of group definition) and we thus set the genome-wide significance threshold at $1 \times 10^{-8}$ ($5 \times 10^{-8}/5$). This work was conducted using the UK Biobank resource (application 19655). Participants enrolled in UK Biobank have signed informed consent forms. Replication analyses for the identified signals associated with food phenotypes were conducted independently by using genetic and dietary data from the EPIC-Norfolk Study [23] and the Fenland Study [24]. Details additional online methods 1.4.

## Investigating the causal effect of health outcomes on reported food intake

Univariable MR analyses were initially conducted to measure the causal effect of health outcomes on food consumption using the TwoSampleMR [25] R package. Exposures of interest were selected amongst those for which nutritional advice is given and included body mass index (BMI), low density lipoprotein cholesterol (LDLc), high density lipoprotein cholesterol (HDLc), Total cholesterol, Triglycerides, Diastolic and Systolic blood pressure, Type 2 diabetes, and coronary artery disease. In addition, we included educational attainment amongst the exposure traits for the multivariable MR, as a proxy of socio-economic status which is likely to affect food consumption. The full list of studies from which the summary statistics were derived is detailed in Table F in S1 Table. For each exposure we selected all SNPs with $p < 5 \times 10^{-8}$ and $r^2 < 0.001$ to be used as instruments in the MR analysis. After performing stepwise heterogeneity pruning to remove SNPs which showed evidence of heterogeneity in the causal effect estimate, we performed MR analysis using the inverse variance method [26]. We then tested if the intercept from the MR-Egger [27] regression was different from zero ($p < 0.05$). If this was the case, MR-Egger was used for the analysis instead.

## Identification of genetic variants with predominantly direct effects on diet

One of the most important assumptions in MR is that the effect of the instrument on the outcome must be mediated only through the exposure of interest (sometimes referred as exclusion restriction criteria) [28]. In this light, genetic instruments whose effect on food is mediated through the health outcomes or through educational attainment may violate this assumption acting as confounders in the relationship between the exposure and the outcome. Moreover, if the mediating trait is acting on the reporting of food consumption and not food consumption itself it would mean that the genetic variant is not truly associated to food consumption, and it would thus not be a valid instrument. It is thus important to estimate the direct effect (i.e., the effect that acts directly on food intake rather than is mediated through other factors see Fig 2) the SNPs are exerting on actual food consumption in order to properly select the genetic variants to be used as instrumental variables.

To this end we use a modified version of bGWAS [29], in which corrected estimates for genetic variants are obtained after accounting for the effect of other phenotypes on

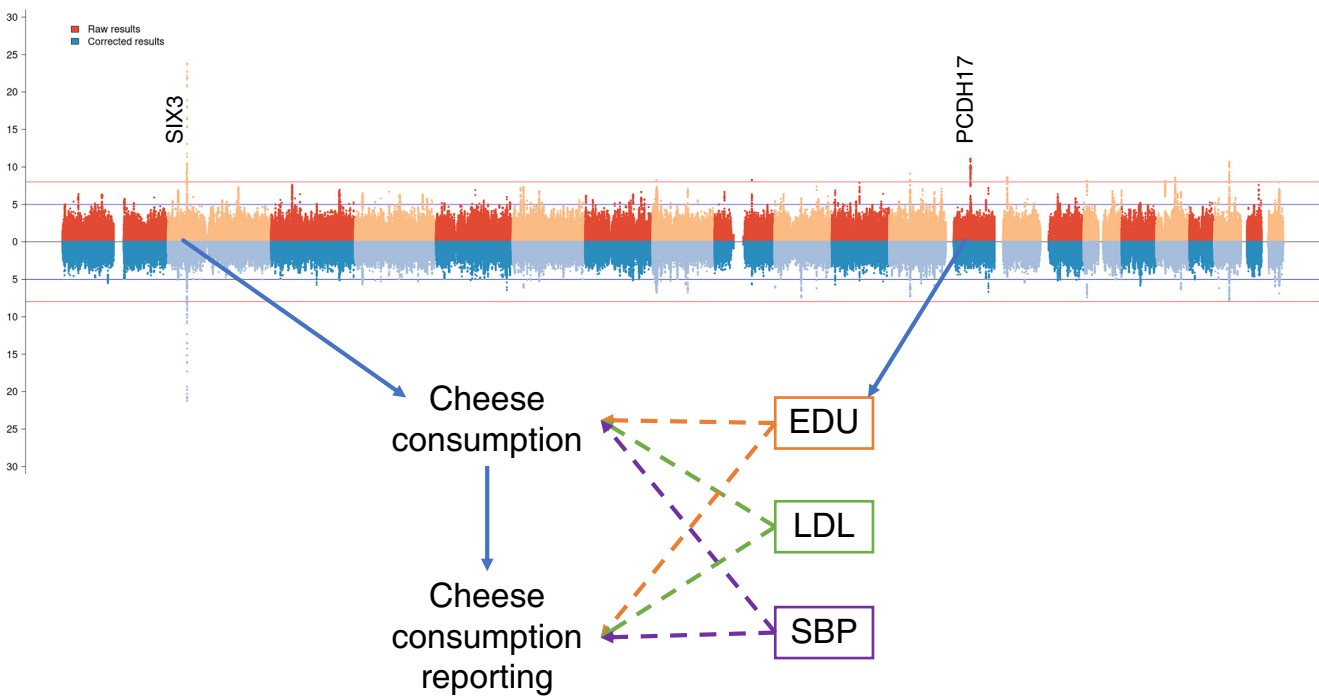

**Fig 2. Direct and indirect SNP effects.** The plot shows the causal path of exemplar genes identified for cheese consumption. In the multivariable MR model cheese consumption is causally influenced by educational attainment (EDU), low density lipoprotein cholesterol levels (LDL) and systolic blood pressure (SBP). The effect of PDCH17 and is mediated through educational attainment, while SIX3 has a direct effect on cheese consumption. The mediated effects cannot be used reliably as MR instruments as they could be affecting either consumption or its reporting. Moreover, they could act as confounders in the MR analysis and thus they need to be identified.

these genetic variants. Further details about bGWAS can be found in S1 Note 1.6. We applied bGWAS to all 29 food phenotypes. As potential mediators, we used the same cardio-metabolic phenotypes as before except total cholesterol to avoid collinearity issues with LDL and HDL cholesterol, and we added summary statistics from Crohn's disease and ulcerative colitis and smoking as they are likely to affect dietary patterns. A full list of the traits used as exposures and their sources can be found in Table F in S1 Table. We identified genetic variants with only a direct effect on diet based on the corrected to uncorrected ratio (CUR) as the ratio between the corrected and the uncorrected effects (see additional methods 1.7 for a detailed explanation). The threshold to define genetic variants with non-mediated effects (CUR = 1±0.05) is based on simulations provided in the S1 Note 2.1 and on the genetic variants with known biological function (i.e. bitter taste receptors). We defined as "non- mediated" those SNPs whose CUR fell within the defined ranges while "uncertain" the others.

## Genome-wide genetic correlations between corrected dietary intake and health outcomes

We used LD-score regression implemented in LD Hub [20,21] to estimate genome-wide genetic correlations between dietary intake phenotypes and 844 health outcomes and intermediary phenotypes. Genetic correlations were estimated both with the corrected and uncorrected GWAS summary statistics using the bivariate LD-score regression model. Stratified

LD-score regression [30] analyses were implemented using ldsc and the annotation files available on the ldsc website.

## Definition of food group variables

In order to define measures of dietary patterns we first performed cluster analysis of the 29 food items applying iCLUST [31] to the corrected genetic correlation matrix between the different foods. iCLUST clusters items in different groups based on a hierarchical structure (Details additional methods 1.8). Fig 3 shows the resulting dendrogram and its comparison with the genetic correlation matrix.

We then defined based on the resulting structure several measures of dietary pattern at different levels of the dendrogram as shown in Fig 3. For each group we performed principal component analysis of the items of that group. The rotation matrix was derived from the eigen decomposition of the genetic correlation matrix of the foods in the PC trait of interest. For example, for the Coffee PC measure we performed principal component analysis of "Ground Coffee", "Instant Coffee" and "Decaf Coffee". Once the rotation matrix was estimated for each SNP its effect on the new measure was estimated as the linear combination of the effect on each food trait using as weights the loadings on each PC. This method has been described before in Tsepilov et al. 2020[32]. For each group of traits only the first component which explained the greatest amount of genetic variance was retained for further analyses. A correlation plot of the loadings of each item onto the PC traits can be found in Fig G in S1 Note.

## MR analyses to assess causal relationships between food intake and health outcomes

MR analyses were conducted to estimate the effects of the food phenotypes on 64 health related phenotypes (see Table S in S1 Table for details) available in MR-base.[25] Genetic instruments for each exposure of interest included independent genetic variants ($p < 5 \times 10^{-8}$ and pruning for LD ($r^2 < 0.001$)). For dietary patterns exposures SNPs were selected as outlined in additional methods 1.9. Briefly once each defined group of traits we estimated the loadings of each index item as the eigen decomposition of the corresponding correlation matrix as outlined in the previous section. This procedure was repeatedly applied to both the original and corrected effects which allowed us to estimate for each SNP on each PC trait effect size, standard error, p-value and CUR. The PC traits where then treated as any other trait applying the same p-value threshold to the projected traits.

For the main analysis we restricted the genetic instruments to those that had evidence of purely direct effect (i.e., not affecting the main exposure through a different pathway; CUR 1 ±0.05). Discussion of the relationship with other methods can be found in S1 Note 2.7. Weights for the genetic instruments were based on the uncorrected effects. To verify the effects of using only direct effect only SNPs on MR, all the analyses were also conducted without applying the CUR filtering.

After selecting the genetic instruments, exposure and outcome data were harmonised. The MR estimates were tested for heterogeneity and outliers were removed using the MR-Radial method.[33] MR analyses were based on the inverse variance weighted method, which estimates the causal effect of an exposure on an outcome by combining ratio estimates using each variant. A random effect model was used if significant heterogeneity between the different estimates was detected. We then tested for the presence of directional pleiotropy using the intercept from the MR-Egger regression. MR median and MR-Raps were used as sensitivity analyses. All results have been made available through an online app (https://npirastu.shinyapps.io/Food_MR/) and can be found in additional Table T in S1 Table.

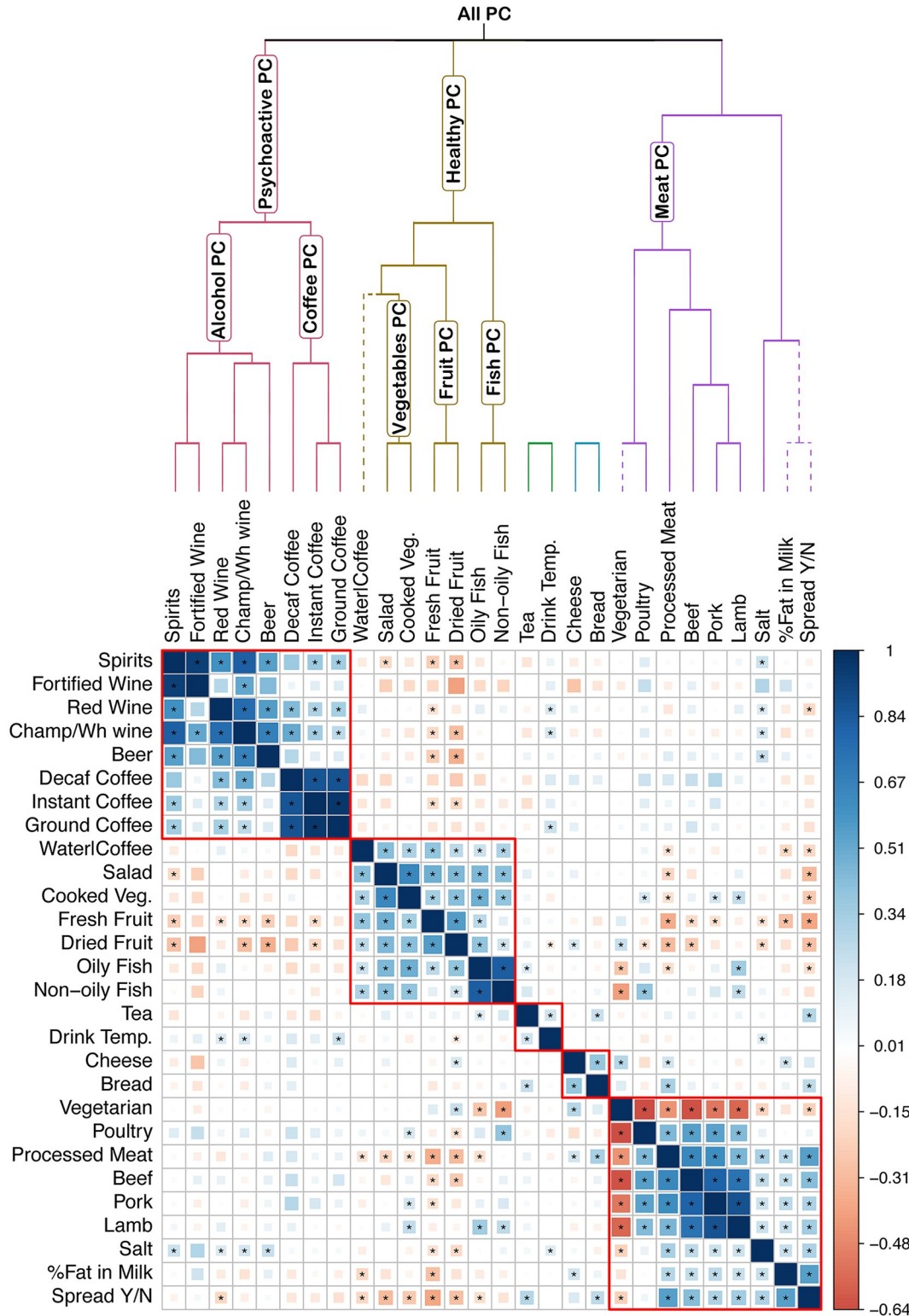

**Fig 3. Clustering of the food traits and definition of measures of dietary patterns.** The plot reports the genetic correlation plot amongst the food traits after applying the correction. The stars report the Bonferroni-corrected significant correlations. The dendrogram and the boxes represent the clustering according to the ICLUST algorithm. The labels on the dendrogram branches show the traits used to define each measure of dietary pattern. The dashed line represents the traits excluded from the estimation of the dietary pattern traits. The "Vegetarian" trait was excluded from the "Meat PC" trait but was included in the overall dietary pattern measure (All PC).

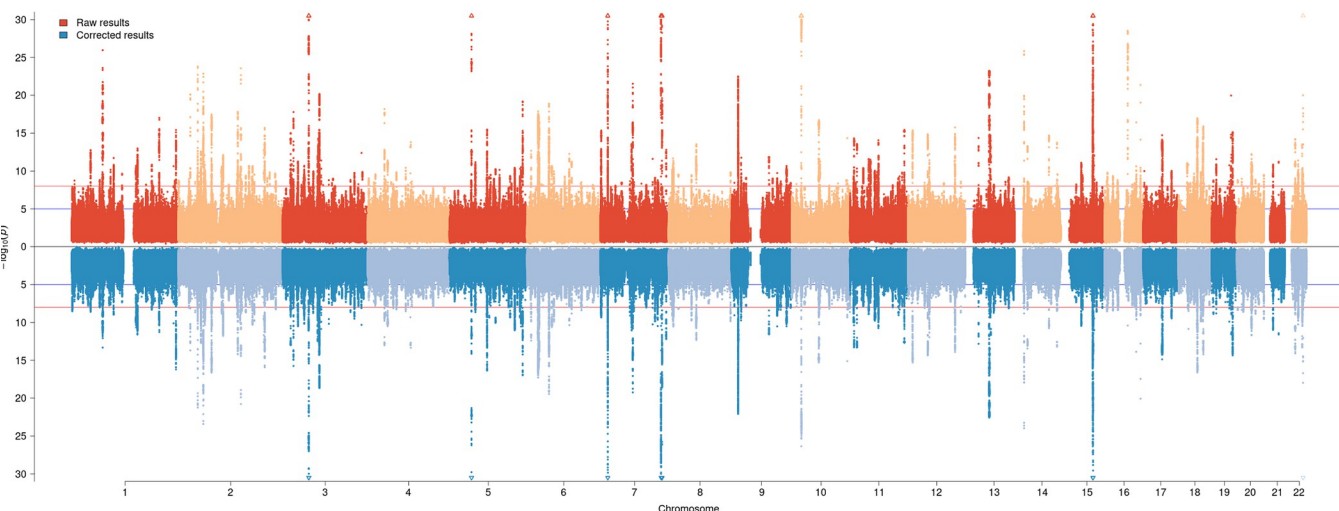

**Fig 4. 302 independent genomic loci associate with food choices.** Results for both univariate (256 loci) and PC traits (additional 27 loci see paragraph S2.3) analyses are included. For each SNP the lowest uncorrected p-value for all traits was plotted. The upper panel represents the unadjusted GWAS associations while the lower panel represents the association with food choices, after adjustment for mediating traits, such as health status for the same snp-trait pair used for the upper panel.

## Results

### Genetic variants associated with food intake

In a GWAS of 29 food phenotypes we identified 412 genetic associations in 256 independent loci (Fig 4 and additional Table D in S1 Table) at Bonferroni corrected level of significance ($P < 1\times10^{-8}$). The principal component association analysis revealed 160 additional SNP-trait associations with additional 27 loci for a total of 572 genetic associations in 283 distinct loci.

Replication was sought in two additional UK-based cohorts including up to 32,779 participants. Despite relatively limited power in replication cohorts, concordant direction of effect was observed for 82% of the signals (p = $7.82\times10^{-35}$, Binomial test; Table E in S1 Table), and nominal significance was achieved by 32% of the signals (p = $9.47\times10^{-54}$). Gene prioritization is described in S1 Note 1.10 while biological annotation, network analysis and tissue enrichment analysis are discussed in additional paragraphs 1.11, 2.4 and 2.5. Several of the identified loci have been previously associated with BMI. However, contrary to our expectations, the BMI-raising allele was consistently associated with lower reported consumption of energy-dense foods such as meat or fat, and higher reported intake of low-calorie foods.

### Genetic variants associated with food intake are strongly influenced by other phenotypes

In univariable MR we identified 81 instances in which health-related traits significantly influencing food intake (Fig 5 additional Table G in S1 Table). For example, BMI and Educational attainment influenced more than 50% of the food traits. Similar effects extend to a broad range of traits, for example LDL and triglycerides influenced 15 and 18 traits respectively.

Higher genetically predicted CAD associates with higher consumption of fish and red wine, and lower consumption of whole milk, salt and lamb. These findings suggest that some of the signals identified in GWAS for reported food phenotypes are not directly associated with food intake but are mediated through a wide range of potential confounders.

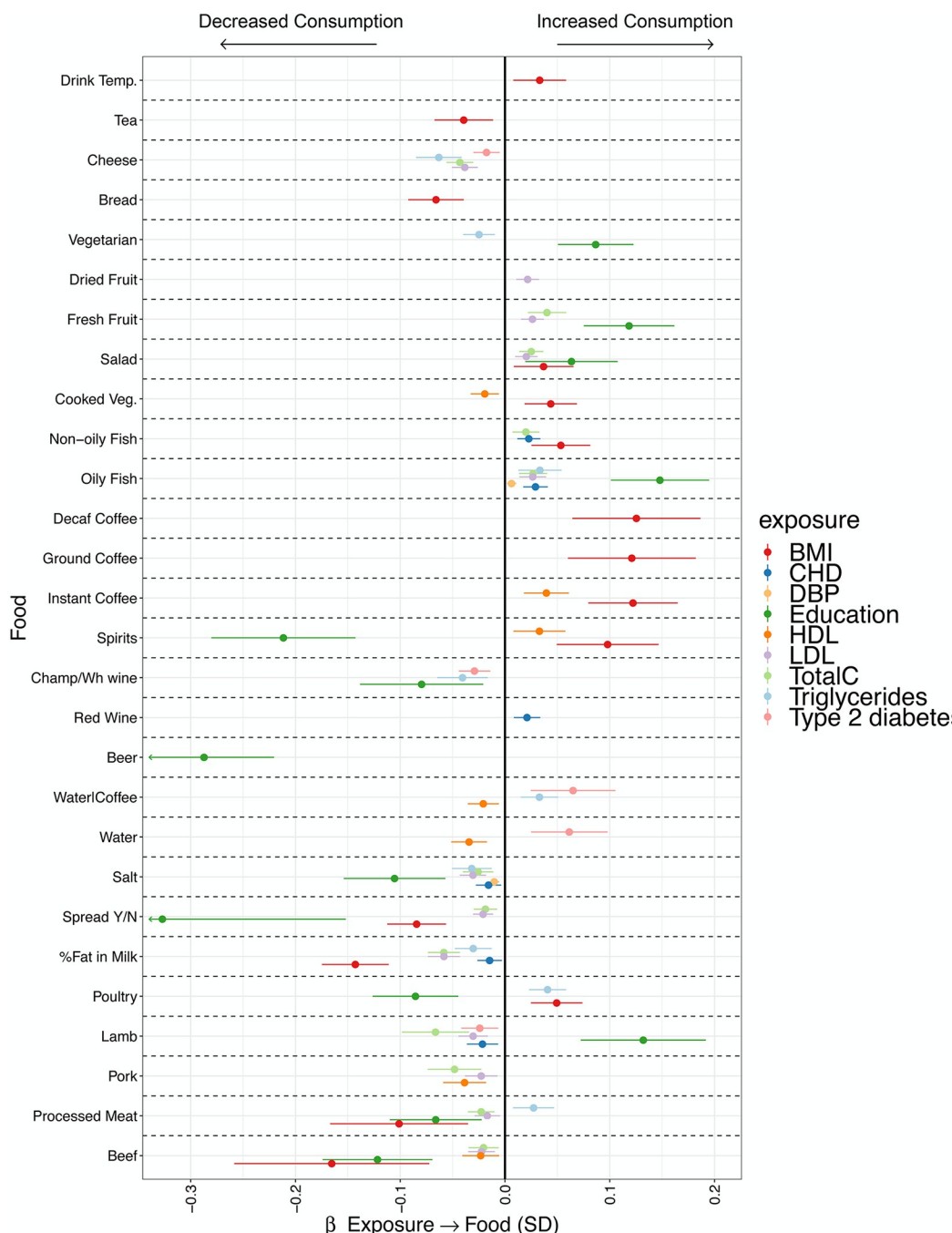

**Fig 5. Health status influences reported food choices.** The plot reports only the univariable MR results which were significant at FDR<0.05. For each food outcome the effect estimate (β) is reported in standard deviations of the exposure trait, together with 95% confidence intervals. Each colour represents a different exposure. BMI, body mass index; CHD, coronary heart disease; DBP, diastolic blood pressure; HDL, high density lipoprotein cholesterol; LDL, low density lipoprotein cholesterol; TotalC, total cholesterol. Champ/Wh wine, champagne, white wine. Temp, temperature.

The Multivariable MR confirmed the univariable MR results (Fig B in S1 Note panel B and Table H in S1 Table). The percentage of genetic variance for the reported food phenotypes explained by health determinants ranged from 42% for cheese to ~0% for fortified wine and white wine/champagne (Fig B in S1 Note panel A and Table H in S1 Table). We systematically

compared the estimated effect sizes of each genetic variants influencing food consumption before and after correcting for the effect of health determinants and showed that in many loci the variant initially identified for food phenotypes changed dramatically after considering the effect of health factors (Fig 4). For example, the effect size of the lead *FTO* variant (rs55872725, $p = 2x10^{-29}$) on milk fat percentage chosen decreased three-fold after accounting for the mediated effects. To further explore the magnitude of this indirect effect on food intake phenotypes, we compared the correlation patterns between the 29 food phenotypes and 832 phenotypes present in the LD hub [21] database identifying great differences. For example, low fat milk intake was correlated with a beneficial effect on body fat percentage ($r_G$ = -0.43) but this association diminished to near zero ($r_G$ = -0.04) after accounting for indirect effects (S1 Note 2.2 and additional Table L in S1 Table). The effects of the correction procedure on the genetic correlation amongst the traits and with the 844 health traits are discussed in S1 Note 2.2 while full results can be found at in Table L in S1 Table and browsed at https://npirastu.shinyapps.io/rg_plotter_2/. These findings highlight the relevance of biases and confounding in genetic correlation studies and we provide a framework to mitigate these problems and to reliably study complex physiological relationships.

## Causal inference analyses for diet phenotypes and health outcomes

A total of 245 out of 572 genetic variants initially associated with food phenotypes were categorized as "non-mediated" associations (Table C in S1 Table). Most loci contained either non-mediated (146/283 loci) or uncertain associations (92/283 loci), while the remaining 45 contained a mixture of the two.

The fraction of uncertain genetic associations varied by food group, ranging from mostly direct effect for tea, decaffeinated coffee, poultry and processed meat, to mostly uncertain for percentage fat in milk and adding spread to bread (Table C in S1 Table).

In two-sample MR analyses we found 52 significant associations between food phenotypes and health outcomes after multiple test correction (q-value < 0.05, Table T in S1 Table). None of them showed sign of heterogeneity amongst the estimates (heterogeneity test q-value >0.05). Fig 6 reports full results for all significant food exposure trait outcome pairs.

Overall, we found that the "overall unhealthy diet" measure did not show significant associations with many traits except for BMI where higher values of this measure corresponded to higher obesity. However, we showed that specific components had effects on different traits. For example, BMI was significantly associated with genetically determined meat consumption, particularly pork and processed meat, but also with a general tendency to heating less healthy foods. In contrast, other measures of adiposity such as waist-to-hip ratio were not associated with meat consumption.

We identified 13 instances in which we would have not detected significant associations without filtering out the non-direct effect instruments such as the effect of increased fruit consumption on triglycerides levels (estimated uncorrected effect = -0.09 (SE = 0.04) vs. estimated corrected effect = -0.17 (SE = 0.05)) or the effect of increased healthy foods consumption on BMI (uncorrected effect = 0.004 (-0.13, 0.14) vs corrected effect = -0.16 (CI 0.07–0.26). In addition, we found 109 food/trait relationships that were not significant after applying CUR filtering, showing that either confounding effects or reduced power explain the lack of association (see additional note 2.6). For example, Psychoactive drinks consumption was initially associated with increased lung cancer (uncorrected effect = 0.27 (CI 0.09–0.45)), but there was little evidence of an association after filtering out the instruments not directly influencing Psychoactive drinks (corrected effect 0.02 (CI -0.17–0.19)). On the flip side, we showed that the effect of alcohol consumption on mean corpuscular volume remains substantially unchanged

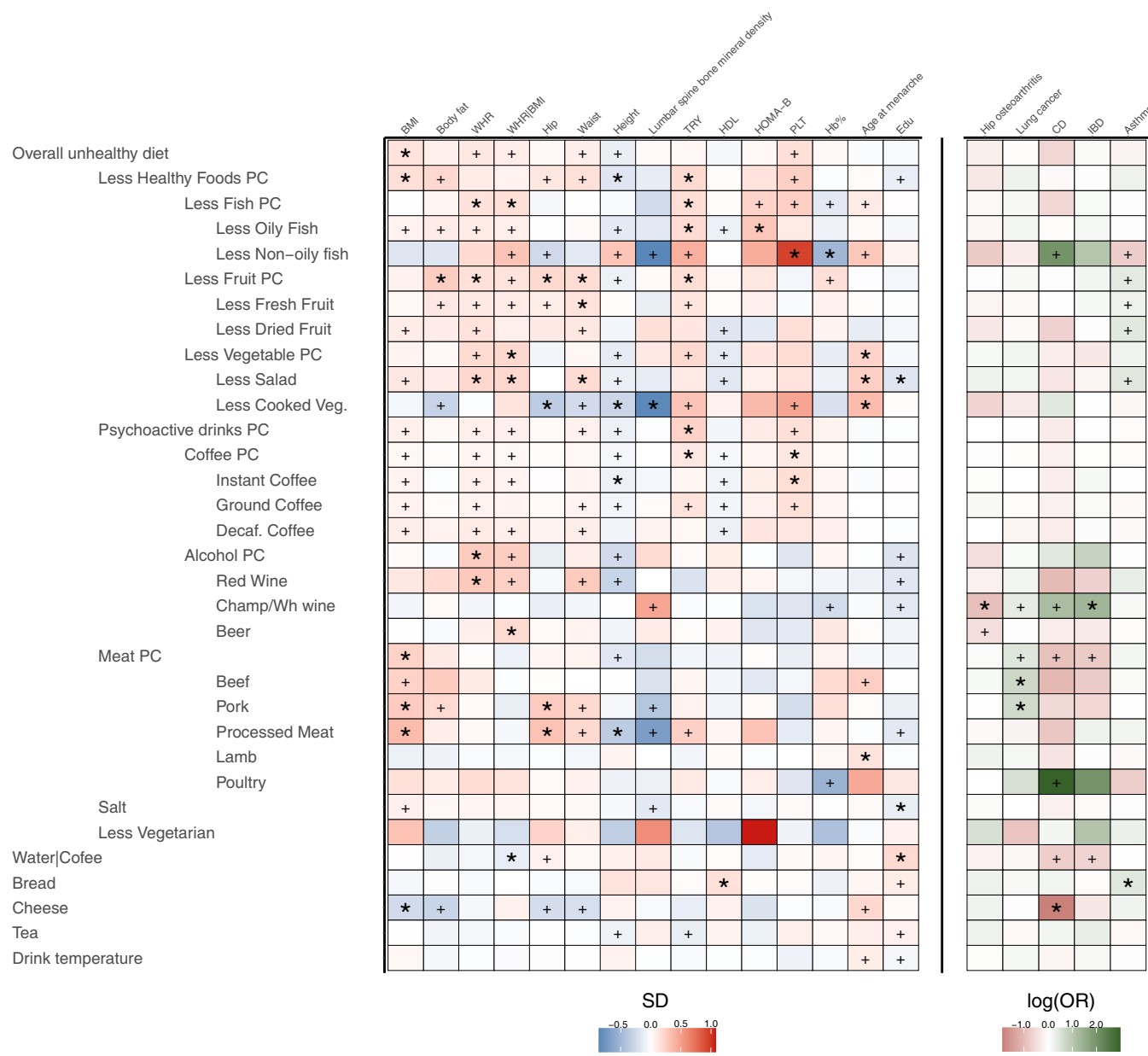

**Fig 6. Significant effects of food choice on disease related traits.** The heatmap reports the results for all significant food trait exposure trait outcome. Only dietary pattern exposures summarising the overall group consumption (PC1) have been reported. All exposures have been aligned to have a positive loading onto the "overall unhealthy diet" measure. Significant food/trait association are indicated with *. Abbreviations: BMI Body Mass Index, WHR Waist to Hip Ratio, TRY triglycerides, TC total cholesterol, HDL HDL cholesterol, LDL LDL cholesterol, Hb% Haemoglobin percentage, PLT Platelet count, Edu Educational attainment, CD Chron's Disease, IBD Inflammatory Bowel Disease. Panel has been divided in two to separate quantitative traits where effect size is in SD$_{outcome}$ per SD$_{exposure}$ (higher effect equals red colour) from qualitative traits where effect sizes are expressed in log(OR$_{outcome}$) per SD$_{exposure}$ (higher effect equals green colour).

when applying the filtering approach (beta 0.05 (SE 0.02) uncorrected and 0.05 (SE 0.06) corrected), suggesting that our approach could precisely identify relevant biological relationships.

A full description of our findings is found in Table T in S1 Table and have been made available through an online app (https://npirastu.shinyapps.io/Food_MR/).

## Discussion

In this study we have quantified the complex interplay between diet and health outcomes showing that the causal path from food intake to adverse health outcomes is not unidirectional and may be influenced by reverse causation and confounding even when MR is used. We showed that genetic correlations and causal inference can be improved by leveraging statistical approaches that consider these mediated effects and identify genetic variants that have predominantly direct effect on the exposure of interest. This information allowed us to perform causal inference analyses that helped identifying more reliable potential causal effects of food on health outcomes.

### Results in context

Previous MR studies have mainly focused on specific food groups such as coffee, alcohol and milk consumption while none has comprehensively investigated the role of different food groups on health outcomes. Here we have expanded this approach to encompass a wide range of specific foods and dietary patterns allowing to compare results across the different traits and giving us more insight in the interpretation of the results. Findings from this study suggest that the biases affecting measures of food consumption (reporting bias, confounding and reverse causation) are propagated to genetic associations. We have shown that these issues extend beyond obesity and socio-economic status, and revealed a broad range of intermediate factors. For example, we showed that LDL and triglycerides concentration influence a wide variety of food traits, implying these phenotypes should be considered as potential sources of bias in future MR studies. For our analyses we have used UK biobank in which participants were aged between 40 and 69 at the time of the questionnaire, it is likely that a younger cohort will suffer less from some of these biases (i.e. altered food consumption due to elevated LDL cholesterol or blood pressure) as it is unlikely that they will display pathological level of these traits.

Our results showing that genetic variations associated with food phenotypes could be influenced by reverse causation and confounding are in contradiction to some previous studies, in which no evidence of reverse causation was reported.[34,35] We believe that this difference is due to our novel approach, which does not correct for potential mediators based on their correlation (through linear regression), but rather based on their causal effect (through MR), which should be able to distinguish the forward and reverse effects when the causal relationship is bidirectional. Our study suggests that it is possible to disentangle these different colliding effects, and identify genetic instruments with a non-mediated effect. This particularity of our approach enables the use of MR for the assessment of causal relationships between food and health.

Many studies have looked at the relationship between nutritional composition and health outcomes. One of the most salient examples is the relationship between saturated fat intake and cardiovascular disease and all-cause mortality, in which recent studies suggest that food sources of saturated fatty acids are more important than saturated fat content per se [36]. Our study provides a new angle on the importance of food sources by showing that foods with similar nutrient profile, for example cheese and meat, have opposite effects on some metabolic risk factors such as BMI but there is no difference in other phenotypes such as blood lipids. Similarly, food with relatively different nutritional composition such as fruit, vegetables, and fish had the beneficial effect on triglycerides. While these findings require further investigations, our genetic evidence lends support for the importance of studying foods in their complexity and not as a mere mixture of nutrients. This approach, in fact, does not consider that the sources of the nutrients are not equal due to the food matrix, the different preparations

and that foods are seldom consumed by themselves but in patterns which are likely to modify the effects on health.

Our findings illustrate that the effect of diet on health outcomes is complex, and components of specific food groups have a differential association with health. In this case, although fish and fruit and vegetables have a very different macronutrient composition it was impossible to separate their effect on triglyceride concentrations. This suggests that at least in this case the macronutrient composition is not as important as an overall tendency to eat certain foods and it highlights the importance of always including the assessment of dietary patterns before claiming health effects of single foods or nutrients. This example also highlights one of the limitations of MR. We can study only specific food exposures for which valid and relatively specific instrument exists and the resolution stops at the point where genes influence food groups/patterns rather than specific items. Better powered studies may enable the identification of genetic markers associated with specific food items, enabling more refined analyses.

Our study also provide evidence that the overall unhealthy diet is almost exclusively associated with BMI, with no evidence of associations with any other outcome. This means that it may be possible to design dietary interventions by modulating only specific food depending on the effect we want to obtain for everyone. For example, if we consider obesity related traits, meat seems to have an effect specifically on BMI while consumption of healthy foods, fish, fruit and vegetables influence fat distribution as indicated by the association with WHR and WHR adjusted for BMI. Although thus there is heterogeneity in effects both lowering BMI and WHR are desirable outcomes and thus and overall healthy diet is still the desirable intervention if we aim at maximising the beneficial health effects across all outcomes.

Our study has several potential limitations. First, the number of items available in the dietary questionnaire in the UK biobank is limited, and therefore it limited our ability to capture overall diet or specific food groups not available. The inclusion of white and relatively healthy and educated participants from UK Biobank may have limited the generalisability of our findings. The self-reported nature of the diet questionnaire is prone to measurement error and bias, and the use of a short food frequency survey could have further reduced the resolution of dietary data collected. More accurate dietary intake assessment methods such as the use of dietary intake biomarkers (doubly labelled water, urinary nitrogen, minerals, and vitamins) for calibration purposes would be valuable in future studies, specially to obtain more precise estimates of the causal effect sizes, however these are challenging to implement in large-scale cohort. Another source of bias may be due to the missing samples due to either nonresponse or to removal due to the phenotype definition which may induce spurious correlation similarly to sampling bias. Moreover estimated effect sizes could be inflated because of the underestimation of the SNP effects on the actual food trait consumption, rather than its self-report, if so, this will have inflated our estimates of the effects of food on health, due to the noise in the questionnaire responses, and warrants further statistical investigations. Even so, our method should not have falsely identified a causal effect or reversed its direction, but further studies are needed to assess the precise effect sizes. Finally, we must consider the possibility of residual confounding effects through variable either not included in our models or imperfectly measured. For example, we have used educational attainment as a proxy for socio economic status, so we cannot exclude that more precise measures would result in even better estimates. Despite these limitations, our methodological approach offers a possibility to improve our understanding of the genetics of diet and strengthen causality in nutrition research.

In conclusion, our findings show that overall, what is generally considered a healthy diet leads to many favourable health outcomes and to reducing a wide range of risk factors broadly agreeing with current guidelines aimed at reducing meat and alcohol consumption while increasing fruit vegetables and fish. We also show that some of these effects are mostly

attributable to specific food or group of foods which however are not characterized by common nutrient composition thus adding granularity to our knowledge on the effect of diet on health. This information can be useful to inform the design and implementation of future studies to reduce the burden of diet-related diseases.

## Supporting information

**S1 Note. Supplementary Methods and results.** Fig A. Directed acyclic graph explaining the two possible scenarios for the effect of a SNP on the trait of interest Y. (a) The SNP has a direct effect on Y not mediated through X. Then the estimated effect of SNP on X will be normally distributed around 0, thus the corrected and uncorrected effects will be similar and their CUR will be close to 1. (b) The SNP effect is mediated through X, thus the corrected effect will deviate from the observed one and CUR will deviate from 1. Fig B. Results for the Mutlivariable MR. Panel A. The plot represents the proportion of genetic variance which is explained by the effect of the health related traits on the food traits. Clearly some of the food traits are extremely biased having up to 25% of genetic variance due to the mediation of the health related traits. Panel B The heatmap represents the effect of the health related traits on each food trait using from the multivariable model. The color is proportional to the effect size. Fig C. Diagram describing the relationships between the simulated traits and their relative parameters. Gy refers to the genetic variants which directly affect Y before any influence of confounding or other mediated traits (Yt). Gu represents the genetic component of a confounder trait U which causally affects both Y and X. Gx represents the genetic component of the outcome trait X which is in turn causally affecting the trait Yi. Yo represents the actual observed trait to which we add noise to reflect the test-retest correlation in FFQ data. Fig D. Scatterplot of the CUR values for Gx (in red), Gy (in green) and Gu (in blue) at the different values of the effect of Y on X and of X on Y. Fig E. Corrected-to-uncorreected ratio (CUR) successfully distinguishes mediated and non-mediated associations. (a) Graph showing mediated and non-mediated pathways. The values of CUR that different types of simulated SNPs (Gx, Gy, Gu) assume at different explained variances ($\sigma^2$) of X->Y when $\sigma$(Y->X)$\neq$0, i.e. presence of reverse causality (b). The values we used for defining a "non-mediated" variant are highlighted in purple. (c) The proportion of variants that are truly Gy, that is directly associated with the trait of interest, across a range of CUR. (d) The overall proportion of variants directly associated with the trait (SNPy) whose CUR falls inside the specified ranges, i.e., the probability of detecting SNPy over all possible scenarios.When the effect of Y->X is equal to zero, Gy is clearly distinguishable from Gx and Gu using CUR (Fig E), however, when $\beta$(Y->X) increases, values of CUR for both Gy and Gx start varying and overlapping (Fig Eb). We thus determined which values of CUR would maximise the probability of correctly selecting Gy under all scenarios. Clearly the parameters we have chosen for defining a "non-mediated" SNPs maximise both the probability of correctly selecting a SNPy. Fig F. Clustering of food consumption traits before and after correction. Comparison between the hierarchical clustering of the food traits based on the uncorrected (on the left) and corrected (on the right) genetic correlations. Black lines connect the same traits for which the clustering has changed. Dendrograms connect the items in each case with the boldness of the line representing the strength of support for the tree nodes. Unique nodes are represented with a dashed line while shared nodes with a bold one. The thickness of the line is thicker for conserved higher level nodes. Fig G. Corrplot of the loadings of each food item onto the measures of dietary pattern. All items have been aligned to the "Overall unhealthy diet" measure. The items that have been flipped are noted as "Less" to clarify the direction of the relationship. Fig H. Heatmap of tissue and functional enrichments. The colour is proportional for the enrichment revealed by stratified LD-score regression. Only

correlations with FDR<0.05 are reported. (a) Enrichment among different classes of functional annotation. (b) Tissue enrichment from Gtex expression. (c) Tissue enrichment from ROADMAP epigenetics. (d) Tissue enrichment from the Franke lab dataset. Fig I. Dotplot of the differential expression analysis run on the prioritised genes from the non-mediated loci and the overrepresentation analysis performed with MAGMA. The overexpressed tissue involved by the two methods were highly overlapping with the analysis performed on the prioritised genes showing the tissues in which there is evidence of underexpression. Fig L. STRING network of genes in non mediated loci. Network plot of the genes in the non-mediated loci. After performing community detection we identified ten different clusters of genes each with its particular set of functions and expression patterns (see additional paragraph 2.6 for details). Nodes have been colored according to community membership. Fig M. Tissues which overexpress the genes in each community. Fig N. Overlap in Go-terms between different communities. The Fig shows that there is no overlap (with the exception of 2 terms) between the terms enriched in each community. The labels have been removed as the plot is meant to only show the overlaps. Fig O-AD show the enriched terms for each community separately. Fig O. Enriched GO-Terms for community 1. Fig P. Enriched GO-Terms for community 2. Fig Q. Enriched GO-Terms for community 3. Fig R. Enriched GO-Terms for community 4. Fig S. Enriched GO-Terms for community 5. Fig T. Enriched GO-Terms for community 6. Fig U. Enriched GO-Terms for community 7. Fig V. Enriched GO-Terms for community 8. Fig Z. Enriched GO-Terms for community 9. Fig AA. Enriched GO-Terms for community 10. Fig AB. Enriched GO-Terms for community 11. Fig AC. Enriched GO-Terms for community 12. Fig AD. Enriched GO-Terms for community 13. Fig AE. Selected forest plots of MR-estimated effect sizes. A Forest plot of the effect of Cheese and Meat consumption on lipid and obesity measures. Despite both foods have a high protein and high fat content their effects on lipid levels and BMI are different. Abbreviations BMI Body Mass Index, TRY Triglycerides, TC Total Cholesterol, LDL Low Density Lipoprotein. B Effect of several foods related to healthy foods on blood tryglicerides levels. Effect for all foods are very similar and make it impossible to distinguish the contribution of each food. Fig AF. Effect of food on obesity related measures. The forest plot compares the effect of each food trait on four obesity related measures: BMI, Body Fat, Waist to Hip Ratio (WHR) and BMI adjusted WHR (WHR|BMI). Each color and shape represents a different obesity related measure. Fig AG. Forest plots of the exposure/outcome pairs significant at the uncorrected analysis. The forest plots represent the estimated effect sizes for all the non CUR filtered MR analyses. The squares represent the point estimates while the bars the 95% confidence intervals. Results from the uncorrected analysis (raw) and CUR filtered IVs (CUR) are reported. The exposure trait is indicated in the header of the plots while the row labels refer to the outcomes. Beta's always refer to standard deviations for the exposure while for the outcomes it is standard deviations for the quantitative traits and log(OR) for the disease traits. Fig AH. The forest plots represent the estimated effect sizes for all the non CUR filtered MR analyses. The squares represent the point estimates while the bars the 95% confidence intervals. Results from the uncorrected analysis (raw) and CUR filtered IVs (CUR) are reported. The exposure trait is indicated in the header of the plots while the row labels refer to the outcomes. Beta's always refer to standard deviations for the exposure while for the outcomes it is standard deviations for the quantitative traits and log(OR) for the disease traits. Fig AI. The forest plots represent the estimated effect sizes for all the non CUR filtered MR analyses. The squares represent the point estimates while the bars the 95% confidence intervals. Results from the uncorrected analysis (raw) and CUR filtered IVs (CUR) are reported. The exposure trait is indicated in the header of the plots while the row labels refer to the outcomes. Beta's always refer to standard deviations for the exposure while for the outcomes it is standard deviations for the quantitative traits and log(OR) for the

disease traits. Fig AL. The forest plots represent the estimated effect sizes for all the non CUR filtered MR analyses. The squares represent the point estimates while the bars the 95% confidence intervals. Results from the uncorrected analysis (raw) and CUR filtered IVs (CUR) are reported. The exposure trait is indicated in the header of the plots while the row labels refer to the outcomes. Beta's always refer to standard deviations for the exposure while for the outcomes it is standard deviations for the quantitative traits and log(OR) for the disease traits. Fig AM. The forest plots represent the estimated effect sizes for all the non CUR filtered MR analyses. The squares represent the point estimates while the bars the 95% confidence intervals. Results from the uncorrected analysis (raw) and CUR filtered IVs (CUR) are reported. The exposure trait is indicated in the header of the plots while the row labels refer to the outcomes. Beta's always refer to standard deviations for the exposure while for the outcomes it is standard deviations for the quantitative traits and log(OR) for the disease traits. Fig AO. The forest plots represent the estimated effect sizes for all the non CUR filtered MR analyses. The squares represent the point estimates while the bars the 95% confidence intervals. Results from the uncorrected analysis (raw) and CUR filtered IVs (CUR) are reported. The exposure trait is indicated in the header of the plots while the row labels refer to the outcomes. Beta's always refer to standard deviations for the exposure while for the outcomes it is standard deviations for the quantitative traits and log(OR) for the disease traits. Fig AP. The forest plots represent the estimated effect sizes for all the non CUR filtered MR analyses. The squares represent the point estimates while the bars the 95% confidence intervals. Results from the uncorrected analysis (raw) and CUR filtered IVs (CUR) are reported. The exposure trait is indicated in the header of the plots while the row labels refer to the outcomes. Beta's always refer to standard deviations for the exposure while for the outcomes it is standard deviations for the quantitative traits and log(OR) for the disease traits. Fig AQ. The forest plots represent the estimated effect sizes for all the non CUR filtered MR analyses. The squares represent the point estimates while the bars the 95% confidence intervals. Results from the uncorrected analysis (raw) and CUR filtered IVs (CUR) are reported. The exposure trait is indicated in the header of the plots while the row labels refer to the outcomes. Beta's always refer to standard deviations for the exposure while for the outcomes it is standard deviations for the quantitative traits and log(OR) for the disease traits. Fig AR. The forest plots represent the estimated effect sizes for all the non CUR filtered MR analyses. The squares represent the point estimates while the bars the 95% confidence intervals. Results from the uncorrected analysis (raw) and CUR filtered IVs (CUR) are reported. The exposure trait is indicated in the header of the plots while the row labels refer to the outcomes. Beta's always refer to standard deviations for the exposure while for the outcomes it is standard deviations for the quantitative traits and log(OR) for the disease traits. Fig AS. The forest plots represent the estimated effect sizes for all the non CUR filtered MR analyses. The squares represent the point estimates while the bars the 95% confidence intervals. Results from the uncorrected analysis (raw) and CUR filtered IVs (CUR) are reported. The exposure trait is indicated in the header of the plots while the row labels refer to the outcomes. Beta's always refer to standard deviations for the exposure while for the outcomes it is standard deviations for the quantitative traits and log(OR) for the disease traits. Fig AT. The forest plots represent the estimated effect sizes for all the non CUR filtered MR analyses. The squares represent the point estimates while the bars the 95% confidence intervals. Results from the uncorrected analysis (raw) and CUR filtered IVs (CUR) are reported. The exposure trait is indicated in the header of the plots while the row labels refer to the outcomes. Beta's always refer to standard deviations for the exposure while for the outcomes it is standard deviations for the quantitative traits and log(OR) for the disease traits. Fig AU. The forest plots represent the estimated effect sizes for all the non CUR filtered MR analyses. The squares represent the point estimates while the bars the 95% confidence intervals. Results from the uncorrected

analysis (raw) and CUR filtered IVs (CUR) are reported. The exposure trait is indicated in the header of the plots while the row labels refer to the outcomes. Beta's always refer to standard deviations for the exposure while for the outcomes it is standard deviations for the quantitative traits and log(OR) for the disease traits. Fig AV. The forest plots represent the estimated effect sizes for all the non CUR filtered MR analyses. The squares represent the point estimates while the bars the 95% confidence intervals. Results from the uncorrected analysis (raw) and CUR filtered IVs (CUR) are reported. The exposure trait is indicated in the header of the plots while the row labels refer to the outcomes. Beta's always refer to standard deviations for the exposure while for the outcomes it is standard deviations for the quantitative traits and log(OR) for the disease traits. Fig AZ. The forest plots represent the estimated effect sizes for all the non CUR filtered MR analyses. The squares represent the point estimates while the bars the 95% confidence intervals. Results from the uncorrected analysis (raw) and CUR filtered IVs (CUR) are reported. The exposure trait is indicated in the header of the plots while the row labels refer to the outcomes. Beta's always refer to standard deviations for the exposure while for the outcomes it is standard deviations for the quantitative traits and log(OR) for the disease traits. Fig BA. The forest plots represent the estimated effect sizes for all the non CUR filtered MR analyses. The squares represent the point estimates while the bars the 95% confidence intervals. Results from the uncorrected analysis (raw) and CUR filtered IVs (CUR) are reported. The exposure trait is indicated in the header of the plots while the row labels refer to the outcomes. Beta's always refer to standard deviations for the exposure while for the outcomes it is standard deviations for the quantitative traits and log(OR) for the disease traits. Fig BB. The forest plots represent the estimated effect sizes for all the non CUR filtered MR analyses. The squares represent the point estimates while the bars the 95% confidence intervals. Results from the uncorrected analysis (raw) and CUR filtered IVs (CUR) are reported. The exposure trait is indicated in the header of the plots while the row labels refer to the outcomes. Beta's always refer to standard deviations for the exposure while for the outcomes it is standard deviations for the quantitative traits and log(OR) for the disease traits. Fig BC. The forest plots represent the estimated effect sizes for all the non CUR filtered MR analyses. The squares represent the point estimates while the bars the 95% confidence intervals. Results from the uncorrected analysis (raw) and CUR filtered IVs (CUR) are reported. The exposure trait is indicated in the header of the plots while the row labels refer to the outcomes. Beta's always refer to standard deviations for the exposure while for the outcomes it is standard deviations for the quantitative traits and log(OR) for the disease traits. Fig BD. The forest plots represent the estimated effect sizes for all the non CUR filtered MR analyses. The squares represent the point estimates while the bars the 95% confidence intervals. Results from the uncorrected analysis (raw) and CUR filtered IVs (CUR) are reported. The exposure trait is indicated in the header of the plots while the row labels refer to the outcomes. Beta's always refer to standard deviations for the exposure while for the outcomes it is standard deviations for the quantitative traits and log(OR) for the disease traits. Fig BE. The forest plots represent the estimated effect sizes for all the non CUR filtered MR analyses. The squares represent the point estimates while the bars the 95% confidence intervals. Results from the uncorrected analysis (raw) and CUR filtered IVs (CUR) are reported. The exposure trait is indicated in the header of the plots while the row labels refer to the outcomes. Beta's always refer to standard deviations for the exposure while for the outcomes it is standard deviations for the quantitative traits and log(OR) for the disease traits. Fig BF. The forest plots represent the estimated effect sizes for all the non CUR filtered MR analyses. The squares represent the point estimates while the bars the 95% confidence intervals. Results from the uncorrected analysis (raw) and CUR filtered IVs (CUR) are reported. The exposure trait is indicated in the header of the plots while the row labels refer to the outcomes. Beta's always refer to standard deviations for the exposure while for the outcomes it is standard

deviations for the quantitative traits and log(OR) for the disease traits. Fig BG. The forest plots represent the estimated effect sizes for all the non CUR filtered MR analyses. The squares represent the point estimates while the bars the 95% confidence intervals. Results from the uncorrected analysis (raw) and CUR filtered IVs (CUR) are reported. The exposure trait is indicated in the header of the plots while the row labels refer to the outcomes. Beta's always refer to standard deviations for the exposure while for the outcomes it is standard deviations for the quantitative traits and log(OR) for the disease traits. Fig BH. The forest plots represent the estimated effect sizes for all the non CUR filtered MR analyses. The squares represent the point estimates while the bars the 95% confidence intervals. Results from the uncorrected analysis (raw) and CUR filtered IVs (CUR) are reported. The exposure trait is indicated in the header of the plots while the row labels refer to the outcomes. Beta's always refer to standard deviations for the exposure while for the outcomes it is standard deviations for the quantitative traits and log(OR) for the disease traits. Fig BI. The forest plots represent the estimated effect sizes for all the non CUR filtered MR analyses. The squares represent the point estimates while the bars the 95% confidence intervals. Results from the uncorrected analysis (raw) and CUR filtered IVs (CUR) are reported. The exposure trait is indicated in the header of the plots while the row labels refer to the outcomes. Beta's always refer to standard deviations for the exposure while for the outcomes it is standard deviations for the quantitative traits and log(OR) for the disease traits. Fig BL. The forest plots represent the estimated effect sizes for all the non CUR filtered MR analyses. The squares represent the point estimates while the bars the 95% confidence intervals. Results from the uncorrected analysis (raw) and CUR filtered IVs (CUR) are reported. The exposure trait is indicated in the header of the plots while the row labels refer to the outcomes. Beta's always refer to standard deviations for the exposure while for the outcomes it is standard deviations for the quantitative traits and log(OR) for the disease traits. Fig BM. The forest plots represent the estimated effect sizes for all the non CUR filtered MR analyses. The squares represent the point estimates while the bars the 95% confidence intervals. Results from the uncorrected analysis (raw) and CUR filtered IVs (CUR) are reported. The exposure trait is indicated in the header of the plots while the row labels refer to the outcomes. Beta's always refer to standard deviations for the exposure while for the outcomes it is standard deviations for the quantitative traits and log(OR) for the disease traits. Fig BN. The forest plots represent the estimated effect sizes for all the non CUR filtered MR analyses. The squares represent the point estimates while the bars the 95% confidence intervals. Results from the uncorrected analysis (raw) and CUR filtered IVs (CUR) are reported. The exposure trait is indicated in the header of the plots while the row labels refer to the outcomes. Beta's always refer to standard deviations for the exposure while for the outcomes it is standard deviations for the quantitative traits and log(OR) for the disease traits. Fig BO. The forest plots represent the estimated effect sizes for all the non CUR filtered MR analyses. The squares represent the point estimates while the bars the 95% confidence intervals. Results from the uncorrected analysis (raw) and CUR filtered IVs (CUR) are reported. The exposure trait is indicated in the header of the plots while the row labels refer to the outcomes. Beta's always refer to standard deviations for the exposure while for the outcomes it is standard deviations for the quantitative traits and log(OR) for the disease traits.
(DOCX)

**S1 Table. Supplementary tables.** Table A: Number of samples used for the GWAS for each trait. The questionnaire column indicates from which questionnaire the item was taken. Proportion of sample indicates the proportion of samples used compared to the number of people who participated in the UK biobank at baseline (501,520). Table B: Description of the phenotypes: the table reports the description and coding of the phenotypes used for the GWAS

analyisis. Phenotype, name of the phenotype; Question, question asked in the tochscreen questionnaire; Answer—Converted to, coding of the phenotype; Covariates, covariates used for the analysis, standard = age+sex; Transformation, transformation applied for normalising the trait. Table C: Significant loci per trait and type. Table D. Summary statistics of top SNP per trait per locus. Locus N, genomic locus number. Chr, chromosome; Start, starting position of the locus; End ending position of the locus; rsid, per trait top SNP in the locus; a1 a0 Effect allele, Other allele; N number of samples; Beta effect of the coded allele; SE Standard error of the uncorrected effect; p p-value of the initial association; Corr.Beta, effect of a1 after the GW mediation correction procedure;Corr.SE Standard error of the corrected effect; Corr.p p-value of the corrected association; CUR corrected to raw ratio; Gene prioritised genes, if the top-SNP was between genes and the closet one was chosen, the distance from the gene was indicated; type category of the association based on the CRR values. Table E. Replication results. The table reports the replication results for the traits/SNP for which replication was availble. Locus.N locus number; rsid SNP use for replication; a0 a1, Other allele effect allele; Beta effect of the effect allele in the discovery cohort; p p-value in the discovery cohort; n number of samples used in the discovery cohort; rep_effect effect of the a1 allele in the replication cohorts; rep_p p-value in the replication cohorts; N_rep, number of samples used for replication. Table F. a: GWAS used for the Multivariate MR analysis and GW mediation analysis. Name, name of the trait; Consortium, consortium that performae the GWAS; Reference link to paper describing the GWAS; Download; source of the summary statistics. b: GWAS used for the Multivariate MR analysis and GW mediation analysis. Name, name of the trait; Consortium, consortium that performae the GWAS; Reference link to paper describing the GWAS; Download; source of the summary statistics. Table G: Full results of univariate MR. Outcome, outcome trait; Exposure, exposure trait. Method used for the MR analysis; nsnp, number of SNPs used for th IV; b effect of the exposure on the outcome; se standard error of the effect; pval, p-value; results_adjusted, Multiple test corrected pvalues. Table H: Multivariable MR results: for each food trait (otucome column) the best model used for the prior estimation is reported. Exposure column indicates the exposure used; Beta the multivariable effect of the exposure on the outcome; Se standard error of the effect; P-value p-value of the effects; $Cor^2$ the squared correlation between the prior and the original z-scores. Table I: Full Genetic correlation amongst food results: The table reports the full genetic correlations between the food traits using both corrected and uncorrected results. p1 trait 1; p2 trait 2; rg genetic correlation; se standard error of the genetic correlation estimate; z z-score of the genetic correlation; p p-value; the extension Raw and Corrected refer to the uncorrected and corrected results respectively. Table L: Genetic correlations between the food traits and common traits. Both uncorrected and corrected (the prefix.Corrected is used to indicate corrected results) results are reported. rg, genetic correlation, se standard error of the genetic correlation, p p.value. Table M: Interaction network results analysis performed with STRING. Table N: Comunity membership of the genes in the STRING interaction network. Table O: GO term enrichment for the 13 Detected communities. Table P. Results for tissue enrichment analysis performed on each community. Only tissues with FDR<0.05 are reported. Table Q. LD score regression intercept and heritability estimates. Table R. Proportion of genetic variance explained of each food trait by the causal health related traits. Trait, name of food trait, rg2 proportion of genetic variance explained, rg2se standard error of the proportion of explained genetic variance. Table S: Information on the outcomes used for MR. Table T. Full results of the Food Mendellian Randomization: The table reports the full resutls of the mendellian randomization of food on all the selected outcomes. Exposure: exposure trait, Outcome: outcome trait; Method: method for the main MR analysis: IVW (FE) inverse variance method fixed effect, IVW (RE) inverse variance method random effect, Wald ratio: Wald ratiomethod; nsnp, number of SNPs

used for the MR; b effect size relative to the method columns; se standard error relative to the method column; pval p-value of relative to the method column; type exposure betas used (Uncorrected, Corrected); outliers number of SNPs excluded because they were foudn to be outliers at the MR radial analysis; egger.pleio p-value relative to the mr egger test for directlional pleiotropy; mr.raps.beta/me.raps.se beta and standard error relative to the MR-RAPS method; beta_egger/se_egger beta and standard error realtive to the MR-Egger analysis; beta_median/se_median beta and standard error realtive to the Median analysis; het.p Heterogeneity test p-value; qval Storeys qvalues, the qvalues have been estimated only on the CRR Uncorrected analysis; sens.het Heterogeneity test run on the estimates coming from the different methods.
(XLSX)

## Acknowledgments

This research has been conducted using the UK Biobank Resource under Application Number 19655. We would like to thank Professor George Davey Smith for the precious feedback, Erin MacDonald-Dunlop, and Pascale Lubbe for help with statistical analyses and Dr. Nana Matoba for providing the results from the smoking GWAS.

We are grateful to all the participants who have been part of the EPIC-Norfolk study, the Fenland study and to the many members of the study teams at the University of Cambridge who have enabled this research.

## Author Contributions

**Conceptualization:** Nicola Pirastu, Eryk J. Grzeszkowiak, Ninon Mounier, Zoltán Kutalik, John R. B. Perry, James F. Wilson.

**Data curation:** Nicola Pirastu, Eryk J. Grzeszkowiak, Fumiaki Imamura, Felix R. Day, Jie Zheng, Nele Taba, Maria Pina Concas, Katherine A. Kentistou, Antonietta Robino, Peter K. Joshi, Krista Fischer, John R. B. Perry, James F. Wilson.

**Formal analysis:** Nicola Pirastu, Ciara McDonnell, Eryk J. Grzeszkowiak, Fumiaki Imamura, Felix R. Day, Jie Zheng, Nele Taba, Maria Pina Concas, Linda Repetto, Katherine A. Kentistou, Antonietta Robino, John R. B. Perry.

**Funding acquisition:** Tõnu Esko, Ken K. Ong, Tom R. Gaunt, Zoltán Kutalik, James F. Wilson.

**Investigation:** Nicola Pirastu, Jordi Merino.

**Methodology:** Nicola Pirastu.

**Project administration:** Nicola Pirastu.

**Resources:** John R. B. Perry, James F. Wilson.

**Software:** Nicola Pirastu, Ninon Mounier, Jie Zheng, Zoltán Kutalik, James F. Wilson.

**Supervision:** Nicola Pirastu, Peter K. Joshi, Krista Fischer, Ken K. Ong, Tom R. Gaunt, Zoltán Kutalik, James F. Wilson.

**Validation:** Fumiaki Imamura.

**Visualization:** Nicola Pirastu.

**Writing – original draft:** Nicola Pirastu, Ciara McDonnell, Ninon Mounier, Jordi Merino, Felix R. Day, Tõnu Esko, Peter K. Joshi, Krista Fischer, Ken K. Ong, Tom R. Gaunt, Zoltán Kutalik, John R. B. Perry, James F. Wilson.

**Writing – review & editing:** Nicola Pirastu, Ciara McDonnell, Jordi Merino, Maria Pina Concas, Zoltán Kutalik, John R. B. Perry, James F. Wilson.

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
