## [Decision Letter · Decision Letter 0]

18 May 2021

Dear Dr Pirastu,

Thank you very much for submitting your Research Article entitled 'Using genetic variation to disentangle the complex relationship between food intake and health outcomes.' to PLOS Genetics.

The manuscript was fully evaluated at the editorial level and by independent peer reviewers. The reviewers appreciated the attention to an important problem, but raised some substantial concerns about the current manuscript. Based on the reviews, we will not be able to accept this version of the manuscript, but we would be willing to review a much-revised version. We cannot, of course, promise publication at that time.

If you decide to revise the manuscript for further consideration at PLOS Genetics, please aim to resubmit within the next 60 days, unless it will take extra time to address the concerns of the reviewers, in which case we would appreciate an expected resubmission date by email to plosgenetics@plos.org.

[LINK]

We are sorry that we cannot be more positive about your manuscript at this stage. Please do not hesitate to contact us if you have any concerns or questions.

Yours sincerely,

Ulrike Peters

Guest Editor

PLOS Genetics

Hua Tang

Section Editor: Natural Variation

PLOS Genetics

We hope the authors find the thorough review by the reviewers helpful. To be considered responsive it will be critical that the authors respond to all comments. The authors need to describe the assessment toll in detail in the methods (including how the 29 food phenotypes have been assessed) so that the reader can better judge the work. The authors should give particular attention to the issue that they start off with an assessment tool that is known to be biased and error prone. This is likely particularly the case for the UKB given its limited dietary assessment tool that is not a food frequency questionnaire. As part of this a comparison with the 24 hr recall assessment is an excellent idea and needs to be added to the paper for being considered responsive to the reviewers’ comments. Accordingly, the authors should tone down their statements about the superiority of their approach compared to other work.

The authors also need to add a comprehensive discussion to the manuscript about the limitations raised by the different reviewers, particularly about the issues of weak instrumental variable bias.

The authors also missed to point out the excellent work conducted to find objective biomarkers for dietary intake, such as doubly labeled water, urinary nitrogen, minerals, and vitamins that can be used in calibration studies.

Why does the # of participants per phenotype varies so dramatically? Does this raising concerns about the reliability of the data?

As pointed out by the reviewers the authors need to reconsider the choices of “exposure” variables or causal health related traits they adjusted in the multivariable MR analysis. Related to this it is not clear how it is possible to investigate the association between foods and BMI if as part of the correction all SNPs that show associations between BMI and the foods were excluded. The same question holds for other “exposure” variables or causal health related traits they adjusted in the multivariable MR analysis.

The paper does not describe well what results are derived from the UKB vs the other 2 UK cohorts included. This needs to be clearly described in the main text (not only in one of the many supplemental sections)

As data on smoking are available the authors should investigate if the association between unhealthy diet and lung cancer is truly confounded by smoking or not.

Reviewer's Responses to Questions

**Comments to the Authors:**

Reviewer #1: The authors write: "we develop a statistical genetics framework that enables us, to directly assess the impact of food choices on health outcomes." No genetic analysis can ever do this, let alone the one implemented in this study.

The authors write in the Abstract that: "that the biases which affect observational studies extend also to GWAS, genetic correlations and causal inference through genetics, which can be corrected by applying our methods." This is not actually demonstrated in the study. The authors simply apply some methods (which are mix of previously described approaches) and generate some results, but we don't know whether they correct any previous biases.

The authors are overly critical of previous studies in the introduction without any supporting citations or solid evidence for their claims, yet then rely on self-reported data themselves.

In essence, the paper boils down to multivariable Mendelian randomization. However, the description is incredibly confusing and the authors seem to constantly mix up mediation and genetic confounding (which cannot be distinguished through statistical analysis alone - biological insight is required).

The choice of outcomes is poorly defined - why not also look at osteoarthritis, stroke, peripheral arterial disease, kidney disease, liver disease etc, which are also affected by dietary choices?

No attempt is made to address weak instrument bias in a multivariable Mendelian randomization setting. This will likely be a major limitation, as the variants explain a small proportion of the variance in the exposures and their associations with different dietary choices are correlated.

Neither does the paper seem to have been properly proofread. For example, on line 426 it says "[citation error]".

Reviewer #2: This is an exciting piece of research that aims to disentangle the association between diet and health outcomes. The authors have build a thorough analytical pipeline to GWAS dietary factors in UK Biobank, replicate findings in independent cohorts and apply novel design ideas within the Mendelian randomization (MR) framework with an aim to achieve unbiased MR estimates for diet and health outcomes. There is dearth of MR findings for dietary intake and health outcomes, as it is difficult to identify robust and non-pleiotropic genetic instruments for dietary intake (except for alcohol, coffee and dairy intake), and the authors have tried to address this limitation of previous research. They should be commended for their approach, but there are some issues I would like to raise that could partially invalidate some of the approaches.

1) The UKBB study has not used a FFQ for dietary assessment, instead there was a touchscreen diet questionnaire offered to the totality of participants. This small but important misconception should be corrected in the Introduction. As a consequence, the likelihood of measurement error in the dietary assessment is higher compared to the use of a more established method of dietary assessment like the FFQ. In addition, only a relatively small range of dietary intakes were assessed with the current questionnaire that might bias the definition of healthy or unhealthy diet pattern that the authors used. Some further discussion on these issues is required. In addition, ~140K participants have completed at least 1 of the 4 web-based 24h dietary assessment post-recruitment, and it would be nice to see a sensitivity analysis with these data.

2) The authors have performed MR analyses of several "exposures" (i.e. BMI, lipids, BP, education, T2D and CHD) in relation to dietary intakes and they have used these results via MR mediation analyses to estimate the association of diet to health outcomes not mediated via one of these exposures in an attempt to limit horizontal pleiotropy. This is a smart idea! However, there are couple of additional exposures that may be important to consider further such as smoking and physical activity. In addition, it was not clear in the manuscript whether the authors have performed bidirectional MR analyses between the dietary intakes and the "exposures". In the presence of a bidirectional association, the approach would likely lead to collider bias or adjust out the complete genetic variants to diet association. More formal evaluation and discussion of the assumptions of mediation analysis is required.

3) The authors used a stepwise pruning for heterogeneity for their definition of instrumental variables, which is an interesting approach but I would be hesitant to base this process only on statistics (i.e. the p-value for heterogeneity) and would like to see biological knowledge of gene function to take a central role in the process. The authors should check the association of their final selected IVs with any secondary traits and comment on residual horizontal pleiotropy.

4) The authors defined locus definition and independence based on physical distance (e.g. 250kb). Did they consider also LD structure in different genomic areas?

5) The final MR results of dietary intakes in relation to health outcomes are impressive, and some but not all associations seem to coincide with the hypothesized directions for the associations. However, most of the associations do not survive the multiple comparison p-value threshold and even less survive the sensitivity MR analyses, which the authors should comment clearer upon in the Results text. Could the authors discuss the potential reasons for this? Does their method lead to lower statistical power or to bias in some circumstances when adjusting for exposures acting as colliders?

Reviewer #3: General comments:

This is an interesting work on a very relevant topic, whether Mendellian Randomization (MR) analysis can help circumvent some of the chronic limitations of nutritional epidemiology when relating the intake(s) of specific foods, as well as patterns of foods, to health outcomes. The study used genetic and dietary data from the UKB, as well as data from several available datasets, to identify genetic variants to be used as instrumental variables in etiological models.

Nevertheless, several important aspects of the current approach are not fully convincing, nor several details of the employed methodology are sufficiently clearly described. As a result, the description of the methodology reads very confusing.

In addition, although the study attempts to address weaknesses of nutritional epidemiology like confounding, reverse causation and exposure measurement errors, within the genetic framework employed, genetic instrumental variables for food intakes are estimates in models where dietary assessments by means of food frequency questionnaires were used. It is well known that these measurements are prone to exposure misclassification, potentially leading to biased identification of relevant genetic variants. These limitations are not even addressed in the current version of the manuscript.

Detailed comments:

Abstract, line 48: “determinants of dietary intake” – what dietary intake, how many food groups, how were they chosen?

Line 50-51: the text reads very optimistic in describing the approach but no details whatsoever are provided to enable the reader understand the novelty and the relevance of the work. In line 53, the text mentions “our work” but again no elements are given to capture the importance of the study.

Overall, the abstract is improvable. It reads rather vague and does not release elements necessary to judge the relevance of the study.

Line 56, the text reads “…and several other risk factors”. Should other morbid condition or health outcomes be mentioned instead?

Line 114: the text is very technical, though some details related to the relevance of the use of the “ldsc” command should be disclosed.

Lines 116-117: the reasoning leading to the choice of using genome-wise (statistical) significance threshold equal to 10xE-8 in the study is not explained clearly.

Lines 120-121: The text does not read appropriate.

Lines 123-124: MR analysis is NOT conducted to measure (estimate) the effect of health outcomes on food consumption. Rather the other way round.

Lines 124-125: “foods for which nutritional advice is given”. This is an important information, but it does not clarify what criteria were used to pick up the chosen list of food items this study focused on.

Line 128, “we included educational attainment” – Where?

Line 131: SNP were included if p<5 x 10E-8 and r2<0.001. Up above the criteria seemed to be 10E-8, not 5 x 10E-8. Also SNP should be selected if r2 was greater than 0.001, not lower.

Lines 139-149: The text does not read correct, nor the MR description is appropriate.

Line 150: what is bGWAS?

Lines 150-159: The description of the MR methodology used in the study is very confusing.

Lines 120-121: The text does not read appropriate throughout page 7. Statements like the ones in lines 152-154 are really obscure. The CUR approach should be referenced. The analysis moved from univariate to multivariable methodology, but te various steps are not clearly outlined.

There is a massive amount of information provided as a supplementary material and Tables. It is not ideal to exhaustively capture several nuances of the study.

Reviewer #4: This is a comprehensive analysis and interesting paper that assesses the utility of using genetic data to unravel the complex relationship between dietary intakes and health outcomes. Some comments:

• Smoking seems like an omission from the analysis as it is well established that smoking influences dietary intakes. The authors included educational attainment as it is likely to affect food consumption. What was the rationale for not including smoking?

• Within the UK Biobank, dietary intake has been assessed 4 times during the follow-up period for a subset of participants. How do the genetic variants related to dietary intake identified using the baseline/enrolment (2006-2010) data associate with self-reported intake during the follow-up period assessments (2012-2013; 2014+; 2019+)?

• The authors mention in the results text that contrary to their expectations, the BMI-raising allele was consistently associated with lower reported consumption of energy-dense foods such as meat or fat, and higher reported intake of low-calorie foods. This observation should be discussed further. Do the authors have a possible explanation for this finding?

• Did the authors consider multivariable MR analyses for the full MR results (Table S18)? One of the key findings was a positive effect estimate for alcohol consumption on lung adenocarcinoma risk. It would be good to see multivariable MR analyses adjusted for lifetime smoking behaviour.

• Why did the authors use education as a proxy of SES when Townsend deprivation index may be a more appropriate indicator?

• Did the authors consider validation of the 2-sample MR results in other studies/populations (e.g., FinnGen, Biobank Japan)?

• The paper although generally clearly written has several grammatical errors and inconsistent formatting throughout. A thorough edit of the paper is required.

**Have all data underlying the figures and results presented in the manuscript been provided?**

Reviewer #1: None

Reviewer #2: Yes

Reviewer #3: Yes

Reviewer #4: Yes

PLOS authors have the option to publish the peer review history of their article (what does this mean?). If published, this will include your full peer review and any attached files.

Reviewer #1: No

Reviewer #2: No

Reviewer #3: No

Reviewer #4: No

---

## [Decision Letter · Decision Letter 1]

20 Jan 2022

Dear Dr Pirastu,

Thank you very much for submitting your Research Article entitled 'Using genetic variation to disentangle the complex relationship between food intake and health outcomes.' to PLOS Genetics.

The manuscript was fully evaluated at the editorial level and by independent peer reviewers. The reviewers appreciated the attention to an important problem, but raised some substantial concerns about the current manuscript. Based on the reviews, we will not be able to accept this version of the manuscript, but we would be willing to review a much-revised version. We cannot, of course, promise publication at that time.

If you decide to revise the manuscript for further consideration at PLOS Genetics, please aim to resubmit within the next 60 days, unless it will take extra time to address the concerns of the reviewers, in which case we would appreciate an expected resubmission date by email to plosgenetics@plos.org.

[LINK]

We are sorry that we cannot be more positive about your manuscript at this stage. Please do not hesitate to contact us if you have any concerns or questions.

Yours sincerely,

Ulrike Peters

Guest Editor

PLOS Genetics

Hua Tang

Section Editor: Natural Variation

PLOS Genetics

A detailed response to the remaining comments from the reviewers is needed.

Furthermore, the response to the first comment of reviewer 3 is not sufficient. Comparing one dietary assessment tool (FFQ) relying on self-report with another dietary assessment tool (24-hr recall) relying on self-report is not proof that the assessment in valid. It would be different if measurements like doubly labeled water or 24 hr urinary nitrogen would be available. Accordingly, it is critical that the authors address these weaknesses and the potential that the exposure misclassification can lead to biased identification of relevant genetic variants and the possible consequences in the discussion. As part of this the authors need to add a discussion on the use of objective biomarkers for dietary intake, such as doubly labelled water, urinary nitrogen, minerals, and vitamins that can be used in calibration studies.

It seems like a strong assumption that the impact of the sizable missingness in the response to the dietary questionnaire is likely modest. This statement needs to be revised and the range of % missing should be added.

Reviewer's Responses to Questions

**Comments to the Authors:**

Reviewer #2: The authors have adequately addressed my comments/suggestions.

Reviewer #4: This is a really interesting study with a clever approach to study the role of dietary intakes on health outcomes. The authors are to be commended for the extensive work undertaken for the revised manuscript. I have the following comments:

• The study design is quite complex. Please could the main text include a figure summarising the different aspects of the study design.

• Why was smoking not considered as a health related trait to be assessed in relation with food intake (while education/SES is)? The rationale for the list of exposures could be better described. Smoking is often associated with unhealthy diet pattern but also with weight loss. Could the authors conduct a sensitivity analysis to examine the effect of smoking on food intake? It is very plausible that smoking would influence the intake of food traits and warrant inclusion.

• One of the authors recently published a nice editorial (PMID: 34980908) on the challenges of using genetics in nutritional epidemiological studies. One of the challenges rightly mentioned is taking into account how dietary intakes can change over the lifecourse. The authors should undertake a sensitivity analysis on this using UK Biobank data. Approximately 49,000 UKB participants had repeat diet assessment during an imaging visit ~8+ years after the baseline information was collected (https://biobank.ndph.ox.ac.uk/showcase/field.cgi?id=1369). How stable/consistent were participants dietary intakes between the baseline and imaging visits? Are a similar pattern of genetic signals found when a GWAS of food intake measured during the imaging visit diet is conducted (n=~49,000 participants)?

• There remain several typos throughout the paper and also in the Shiny app. For example, line 72 – “Mendellian”; line 91 – “Hell et al”; throughout – words capitalized mid-sentence. Please ensure that a careful editorial review is undertaken.

**Have all data underlying the figures and results presented in the manuscript been provided?**

Reviewer #2: None

Reviewer #4: Yes

PLOS authors have the option to publish the peer review history of their article (what does this mean?). If published, this will include your full peer review and any attached files.

Reviewer #2: No

Reviewer #4: No

---

## [Editor Report · Decision Letter 2]

21 Feb 2022

Dear Dr Pirastu,

Thank you very much for submitting your Research Article entitled 'Using genetic variation to disentangle the complex relationship between food intake and health outcomes.' to PLOS Genetics.

The manuscript was fully evaluated at the editorial level and by independent peer reviewers. The reviewers appreciated the attention to an important topic but identified some concerns that we ask you address in a revised manuscript

We therefore ask you to modify the manuscript according to the review recommendations. Your revisions should address the specific points made by each reviewer.

[LINK]

Yours sincerely,

Ulrike Peters

Guest Editor

PLOS Genetics

Hua Tang

Section Editor: Human Variation

PLOS Genetics

It is not clear why the authors only described missingness for alcohol and coffee intake. To be responsive to the editor's previous comment please add in a supplemental table the % missing for each dietary variable that was assessed and refer to this table in the methods or result section so that readers can judge for themselves about the potential magnitude of this problem. The very short and vague sentence in the discussion about "informative missingness" should also refer to this table.

---

## [Editor Report · Decision Letter 3]

8 Mar 2022

Dear Dr Pirastu,

Thank you very much for submitting your Research Article entitled 'Using genetic variation to disentangle the complex relationship between food intake and health outcomes.' to PLOS Genetics.

The manuscript was fully evaluated at the editorial level and by independent peer reviewers. The reviewers appreciated the attention to an important topic but identified some concerns that we ask you address in a revised manuscript

We therefore ask you to modify the manuscript according to the review recommendations. Your revisions should address the specific points made by each reviewer.

[LINK]

Yours sincerely,

Ulrike Peters

Guest Editor

PLOS Genetics

Hua Tang

Section Editor: Human Variation

PLOS Genetics

The authors state that they “have added the proportion of samples used compared to the number of people who have responded to the questionnaire.” It is not clear where this information was added. The table that shows these results need to be specified in the paper and the table needs to be added to the main text or supplement. Furthermore, the authors should add the % missing not only compared to the number of people who have responded to the questionnaire but who was asked to respond to the questionnaire as this is the truly critical denominator to judge missingness.

The authors added “Finally, we have excluded people who reported eating certain foods (e.g. beef) less than once a week due to the very large range of different consumptions which this response corresponds to” It is not clear that this is a good approach. It is very unusual to exclude an entire response group from the analysis as this will limited the range of exposure, which in this case is even not that large as it goes from <1 per week to 0 times per week. There are many good approaches to deal with these type of situations, such as evaluate national dietary assessments to estimate the average intake in this group (<1/wk) and evaluate factors that impact these, such as race, ethnicity, sex, BMI,… so that a value can be assigned within population substrata.

The authors added Line 481: “Another source of bias may be due to the missing samples due to either nonresponse or to removal due to the phenotype definition which may induce spurious correlation similarly to sampling bias. Our method however being agnostic to the source of bias, should be able to include these bias and thus select out the SNPs which are associated because of these biases.”

What evidence exist to make the broad claim that the 2nd sentence is true? It seems that this sentence should be deleted.

---

## [Editor Report · Decision Letter 4]

22 Mar 2022

Dear Dr Pirastu,

We are pleased to inform you that your manuscript entitled "Using genetic variation to disentangle the complex relationship between food intake and health outcomes." has been editorially accepted for publication in PLOS Genetics. Congratulations!

Yours sincerely,

Ulrike Peters

Guest Editor

PLOS Genetics

Hua Tang

Section Editor: Human Variation

PLOS Genetics

Comments from the reviewers (if applicable):

**Data Deposition**

http://datadryad.org/submit?journalID=pgenetics&manu=PGENETICS-D-21-00378R4

**Press Queries**

---

## [Editor Report · Acceptance letter]

3 May 2022

PGENETICS-D-21-00378R4 

Using genetic variation to disentangle the complex relationship between food intake and health outcomes. 

Dear Dr Pirastu, 

We are pleased to inform you that your manuscript entitled "Using genetic variation to disentangle the complex relationship between food intake and health outcomes." has been formally accepted for publication in PLOS Genetics! Your manuscript is now with our production department and you will be notified of the publication date in due course.

With kind regards,

Zita Barta

PLOS Genetics

On behalf of:
